# Drug-Resistant Tuberculosis Treatment Recommendation, and Multi-Class Tuberculosis Detection and Classification Using Ensemble Deep Learning-Based System

**DOI:** 10.3390/ph16010013

**Published:** 2022-12-22

**Authors:** Chutinun Prasitpuriprecha, Sirima Suvarnakuta Jantama, Thanawadee Preeprem, Rapeepan Pitakaso, Thanatkij Srichok, Surajet Khonjun, Nantawatana Weerayuth, Sarayut Gonwirat, Prem Enkvetchakul, Chutchai Kaewta, Natthapong Nanthasamroeng

**Affiliations:** 1Department of Biopharmacy, Faculty of Pharmaceutical Sciences, Ubon Ratchathani University, Ubon Ratchathani 34190, Thailand; 2Department of Industrial Engineering, Faculty of Engineering, Ubon Ratchathani University, Ubon Ratchathani 34190, Thailand; 3Department of Mechanical Engineering, Faculty of Engineering, Ubon Ratchathani University, Ubon Ratchathani 34190, Thailand; 4Department of Computer Engineering and Automation, Faculty of Engineering and Industrial Technology, Kalasin University, Kalasin 46000, Thailand; 5Department of Information Technology, Faculty of Science, Buriram University, Buriram 31000, Thailand; 6Department of Computer Science, Faculty of Computer Science, Ubon Ratchathani Rajabhat University, Ubon Ratchathani 34000, Thailand; 7Department of Engineering Technology, Faculty of Industrial Technology, Ubon Ratchathani Rajabhat University, Ubon Ratchathani 34000, Thailand

**Keywords:** ensemble deep learning, multiclass-AMIS, chest X-ray, drug-resistant, tuberculosis, web application diagnosis

## Abstract

This research develops the TB/non-TB detection and drug-resistant categorization diagnosis decision support system (TB-DRC-DSS). The model is capable of detecting both TB-negative and TB-positive samples, as well as classifying drug-resistant strains and also providing treatment recommendations. The model is developed using a deep learning ensemble model with the various CNN architectures. These architectures include EfficientNetB7, mobileNetV2, and Dense-Net121. The models are heterogeneously assembled to create an effective model for TB-DRC-DSS, utilizing effective image segmentation, augmentation, and decision fusion techniques to improve the classification efficacy of the current model. The web program serves as the platform for determining if a patient is positive or negative for tuberculosis and classifying several types of drug resistance. The constructed model is evaluated and compared to current methods described in the literature. The proposed model was assessed using two datasets of chest X-ray (CXR) images collected from the references. This collection of datasets includes the Portal dataset, the Montgomery County dataset, the Shenzhen dataset, and the Kaggle dataset. Seven thousand and eight images exist across all datasets. The dataset was divided into two subsets: the training dataset (80%) and the test dataset (20%). The computational result revealed that the classification accuracy of DS-TB against DR-TB has improved by an average of 43.3% compared to other methods. The categorization between DS-TB and MDR-TB, DS-TB and XDR-TB, and MDR-TB and XDR-TB was more accurate than with other methods by an average of 28.1%, 6.2%, and 9.4%, respectively. The accuracy of the embedded multiclass model in the web application is 92.6% when evaluated with the test dataset, but 92.8% when evaluated with a random subset selected from the aggregate dataset. In conclusion, 31 medical staff members have evaluated and utilized the online application, and the final user preference score for the web application is 9.52 out of a possible 10.

## 1. Introduction

Tuberculosis (TB) predominantly affects the lungs and can have catastrophic effects. Droplets released from the respiratory system while coughing and sneezing are the major mechanism of transmitting the bacteria that causes tuberculosis [1]. Most instances of tuberculosis can be cured with the proper therapy; however, untreated individuals are still at risk of death. When bacteria acquire resistance to the antibiotics used to treat tuberculosis, a type of disease known as drug-resistant tuberculosis can emerge. This shows that the tuberculosis bacteria have developed resistance to treatment [2].

A person Is more likely to contract drug-resistant tuberculosis if they: fail to take their anti-TB medications as prescribed, develop the TB disease again after previous treatment, originate from a region where drug-resistant TB is prevalent, or have close contact with someone who has it [3]. To prevent the spread of drug-resistant tuberculosis (DR-TB), it is essential to take all TB treatments exactly as prescribed by a physician. There should be no missed doses, and treatment should not be discontinued too soon [4]. Individuals undergoing treatment for tuberculosis should notify their healthcare provider if they are having trouble taking their prescriptions. Health care professionals can contribute to the prevention of DR-TB by diagnosing cases immediately according to approved treatment protocols, monitoring patients’ responses to therapy, and ensuring that therapy is completed. Rapid classification and detection help to prevent the spread of tuberculosis and drug-resistant tuberculosis, and simplify treatment.

The tuberculin skin test (TST) and sputum test are the two most important diagnostic instruments for tuberculosis (TB) [5]. Positive results on either the tuberculosis skin test or the tuberculosis blood test indicate that the TB bacteria have been present in the body. This test is unable to distinguish between latent tuberculosis infection (LTBI) and current TB illness. Additional tests, such as a chest X-ray [6] and sputum analysis [7], are required to diagnose tuberculosis. Using specialized laboratory techniques, the drug sensitivity of microorganisms or the types of drug resistance can be determined. Due to a shortage of medical equipment, this type of testing may take considerable time in some medical centers.

Currently, a chest X-ray (CXR) is frequently used to diagnose tuberculosis (TB) and to categorize drug-resistant forms. According to Iqbal, A. et al. [8], TBXNet is a straightforward and efficient deep learning network that can successfully classify a large number of TB CXR images. The network consists of five dual convolution blocks with filter widths of 32, 64, 128, and 256 bits, respectively. The dual convolution blocks are coupled with a learned layer in the network’s fusion layer. Moreover, the pre-trained layer is used to transfer pre-trained knowledge into the fusion layer. On dataset A, dataset B, and dataset C, respectively, the suggested TBXNet has obtained a precision of 98.98%, 99.17%, and 95.10%. Regarding the testing and diagnosis of drug resistance via CXR, Karki et al. [9] proposed a deep learning technique for categorizing DS-TB and DR-TB. The CNN architecture ResNet18 was utilized in this investigation. Tulo et al. [10] classified four unique forms of drug-resistant TB using machine learning. This comprises drug-sensitive tuberculosis (DS-TB), drug-resistant tuberculosis (DR-TB), multidrug-resistant tuberculosis (MDR-TB), and extensively drug-resistant tuberculosis (XDR-TB). Each pair of drug-resistant types has been categorized using a binary classification model capable of predicting two classes at each prediction interval. In Ureta and Shrestha [11], Tulo et al. [12], Jaeger et al. [13] and Kovalev et al. [14], the binary classification approach has also been employed to categorize various pairs of drug-resistant TB.

Deep learning and machine learning have been effectively utilized to identify tuberculosis (TB) and to classify the patient’s drug-resistance type in order to propose the most effective treatment plan to limit the spread of the bacteria. This is due to the fact that there is no previously published model capable of diagnosing tuberculosis and classifying the type of medication resistance in a single model. In this work, we will develop a model that can determine whether a patient has tuberculosis, and if so, which treatment plan should be selected. Therefore, the contribution of our proposed model is as follows.

It is the first model that can determine from CXR image whether a patient has tuberculosis and, if so, what kinds of drug resistance they may have.The proposed model is the first to employ ensemble deep learning to make decisions in multiclass classifications for tuberculosis and drug-resistance classification.For the users, we design computer applications. It is the first application that can determine whether a patient has tuberculosis and, if so, what type of drug resistance they have, using only a CXR image.In the application, the recommended regimen that is appropriate for a specific patient will be displayed.

The organization of the following sections is as follows: The Section 1 will review related work, the Section 2 and Section 3 will present the computational results and discussion, and the Section 4 will describe the materials and methods employed in this study. Section 5 will contain a report of the investigation’s conclusion.

### 1.1. Related Work

The related literature will be separated into three subsections: (1) artificial intelligence (AI) for TB detection, (2) AI for drug-resistant categorization, and (3) web applications in health diagnosis.

#### 1.1.1. AI for TB Detection

In the literature, numerous classic deep learning networks for accurately identifying tuberculous and healthy CXR pictures are proposed. The following is a brief summary of the scientific contributions made to the TB research society. Chest radiograph images were classified by Satyavratan et al. [15] using the Bag of Features technique and the SURF descriptor to classify TB. In addition, a Distance Regularized Level Set is employed for lung field segmentation. The multilayer perceptron network is used as a classifier to distinguish between TB and non-TB images. Researchers have suggested an automatic recognition method for lung cavity sign detection utilizing CT scan pictures in Han et al. [16]. A two-stage, three-stream convolutional neural network (TSCNN) was proposed for detecting lung nodules [17]. The accuracy and F-measure of the proposed model are 80.67% and 78.79%, respectively. First, lung nodule detection is accomplished using U-Net, a segmentation network. Step two involves using a three-tree deep convolutional neural network (DCNN) classification based on a TSCNN with a dual-poling structure to cut down on false positives. Using microscopic images, Momeny et al. [18] suggested a CNN technique for classifying Mycobacterium tuberculosis. Using mixed poling instead of baseline and dropout improves the generalization, and PReLu has improved classification accuracy. The model proposed by Momeny et al. [18] has an accuracy of 93.4%. Lu et al. [19] presented TBNet, a context-aware graph neural network for TB diagnosis. As a backbone network, EfficientNet is used to create sample-level features. The context-aware feature exaction block is utilized on the feature space’s nearest neighbors. Finally, the CARFLN network is trained as a tuberculosis detection classifier. Momeny et al. [18] present a model whose accuracy and F-measure are 98.93% and 98.91%, respectively.

In order to extract characteristics from X-ray pictures, Rahman et al. [20] employed three pre-trained neural networks: VGG19, ResNet101, and DenseNet-201. To categorize X-ray pictures as tuberculosis or normal, the XGBoost network is implemented. Iqbal et al. [21] proposed a dual mechanism for attention that is a mix of lesion-based and channel-based attention. Additionally, a multiscale fusion block is developed to extract richer and more diversified image information. Tasci et al. [22] propose a voting-based enable learning approach employing pre-processing and data augmentation using images of tuberculosis. To fine-tune the CNN’s models, a number of changes are applied. Using a combination of weighted and unweighted voting, the optimal performance is achieved. In addition, the suggested voting method can be used to address additional image recognition issues, including object recognition, disease categorization, etc. The model proposed by Tasci et al. [22] has an accuracy and F-measure of 98.85% and 98.86%, respectively.

Images of tuberculosis, pneumonia, and normal CXRs were classified using reinforcement learning in Kukker, A. and Sharma, R. [23]. The fuzzy Q learning method is used with wavelet-based preprocessing to create a classifier for determining the severity of tuberculosis and pneumonia, and an average accuracy of 96.75% was obtained. Khatibi et al. [24] provide multi-instance, complex network, and stacked ensemble networks for accurately detecting X-ray pictures of tuberculosis and normal tissue. In addition, several patches are obtained, and then the feature vector is cauterized. Additionally, all of the obtained patches are clustered globally, assigning purity scores to each image and aggregating them into the corresponding cluster. According to Khatibi et al. [24], the accuracy is 96.80%.

#### 1.1.2. AI in Drug-Resistant Classification

Karki et al. [9] suggested a technique of deep learning for classifying DS-TB and DR-TB. This research utilizes the CNN architecture ResNet18. Furthermore, potentially confusing objects are removed, and the lungs are downsized to fill the same proportion of the image. Lung size and patient positioning, for instance, are frequently associated with clinical situations and could function as confounding variables. Upon completion of the segmentation, the results demonstrate that incorporating these techniques into their suggested model is highly beneficial. The accuracy and area under the curve (AUC) of the model proposed by Karki et al. [9] are 72.00% and 79.00%, respectively.

Radionics or electromagnetic therapy has been used to detect patterns that radiologists may have missed when investigating anomalies in medical images. Multi-task learning was used to standardize the basic model and to direct the network’s attention to the ostensibly significant areas of the image [19]. The original CXR is sent to U-Net, which develops a binary lung mask in order to prepare the CXR picture for processing. The original CXR is then cropped, and the lungs are segmented inside the decreased bounding box. Tulo et al. [10] utilize machine learning (ML) to assess the lungs and mediastinum in order to categorize the medication response of tuberculosis (TB) patients. Other types of drug-resistant tuberculosis, DS-TB, MDR-TB, and XDR-TB, are predicted despite the use of binary categorization [25] in this investigation. Several machine learning methods, such as multi-layer perceptron [26], K-nearest Neighbor [27], and support vector machine [13], were utilized to determine the solution. The model proposed by [13,26,27] has an accuracy of 98.14%, 93.37%, and 62.00%, respectively.

Caseneuve et al. [28] suggested two algorithms referred to as the threshold approach and edge detection, to identify CXR images with two easily identifiable lung lobes. This model has an accuracy of 97.16%. The notions of Otsu’s threshold [29] and the Sobel/Scharr operator [30,31] served as the basis for these two procedures. Sobel examines the presence of upper/lower image borders or left/right image edges in horizontal and vertical gradient computations. In other words, the Sobel operator will tell whether the visual changes occur quickly or gradually. The Scharr operator provides a variation in which the mean squared angular error is calculated as an additional optimization and the provision of enhanced sensitivity. This permits kernels to be of a derivative type, which approximates Gaussian filters more closely.

In order to improve the quality of the images before feeding them into the CNN model, Ahamed et al. [32] and Wang et al. [33] used image processing techniques such as cropping and sharpening filters to apply to the collected datasets. This was performed in order to improve the image quality by isolating the primary part of the image of the lungs, and also eliminating undesired or superfluous parts of the photographs. The images were cropped to achieve the optimal height-to-width ratio before being uploaded. After the photos were acquired, a sharpening filter was applied to each one of them to improve the quality. This filter is an illustration of a second-order or a second-derivative system of augmentation that emphasizes areas with rapid intensity variation [34]. The AUC of the model ranges from 66.00% to 93.50%, the F-measure ranges from 61.00% to 93.60%, and the accuracy ranges from 62.00% to 87.00%, based on our current evaluation of the relevant literature.

#### 1.1.3. Web Applications in Health Diagnosis

Recent advancements in web technology and browser characteristics have allowed for the creation of interactive medical data solutions on the internet, which in turn has improved healthcare delivery for the general populace [35]. Further, volume rendering on the web is an essential tool to visualize 3D medical datasets and to provide crucial information about the spatial relations between distinct tissue structures [36]. Through the use of web-based volumetric data visualization, medical professionals from any location with access to the internet can work together to diagnose diseases and to schedule surgeries in advance [37].

Due to developments in technology and healthcare systems, the significance of web-based data visualization, particularly in medicine, is growing. There are numerous intriguing studies and uses of web-based medical data visualization; for instance, Congote et al. [38] proposed a ray casting methodology built using WebGL and HTML5, which was based on 2D texture composition and a post-color categorization approach. Mobeen and Feng [39] created a single-pass rendering process for smart phones by simulating 3D textures using 2D textures for volume rendering.

The software packages developed by Mahmoudi et al. [40] and by Marion and Jomier [41] for viewing 3D medical datasets online use WebGL and WebSocket, respectively, and applying this to collaborative data exploration, respectively, leverages AJAX technology in web user interfaces. Rego and Koes [42] employed WebGL and JavaScript to exhibit molecular structures online, and Jaworski et al. [43] created a ray casting platform to display microlevel structural models of composite materials online. Similarly, Sherif et al. [44] used WebGL-based ray casting to show protein structures, and Yuan et al. [45] used the same technique to display macromolecular structures in the context of the computational development of drugs. Virtual reality (VR) was incorporated into a web-based medical visualization framework by Kokelj et al. [46] so that users may investigate medical data in a VR setting. The uses of WebGL, JavaScript, and HTML5 have also been applied to the visualization of a 3D surface and volumetric neuroimaging data in up-to-date web browsers [43], the dynamic display and analysis of the genome on the internet [47], the interactive visualization and analysis of the 3D fractal dimension of MRI data on the web [48], and the interactive browsing and segmentation of medical images on remote client machines [49].

## 2. Results

The computational findings will be presented in this section, broken down into two sub-sections: (1) the test to demonstrate the efficacy of the proposed methodologies, and (2) the outcome of the process of constructing a web application for the classification of drug reaction and the TB detection. The proposed model source code is available at https://github.com/aiosmartlab/AMIS-Ensemble-DL (accessed on 9 November 2022).

### 2.1. A Test for the Effectiveness of the Proposed Methods

The test will be utilized to determine which of the combinations that are presented in Table 3 are the most effective strategy for identifying tuberculosis, as well as the type of drug resistance exhibited by the patients. The efficiency of the proposed model will then be compared to those of the heuristics that are already in place in Section 2.1.2.

#### 2.1.1. Revealing the Most Effective Proposed Methods

All 12 proposed models will be examined for their classification qualities across several classes, using a five-class classification system, and the results of all 12 techniques are displayed in Table 1. We separated the total dataset into a training set and a test set with proportions of 80% and 20%, having 5606 training images and 1408 test images. Consequently, the training set was tested utilizing five-fold cross-validation [50,51]. The test set served as a separate holdover for final evaluation.

The experiment is conducted with batch sizes of 16, 2, and 8 for MobileNetV2, EfficientNetB7, and DenseNet121, respectively. The number of epochs utilized to train the model is 200, and the learning rate using the Adam algorithm, as suggested in Kingma and Ba [52].

The results in Table 1 demonstrate that the N-12 achieved the highest performance rating of all key performance indicators. It provides a 92.6% accuracy rate across all test datasets. It can increase the quality of solutions from N-1, a typical ensemble deep learning model, by 43.56 percent. The N-12 model, which comprises image segmentation, image augmentation, and the use of AMIS-WLCE, also performed the best among all models when subjected to 5-fold cross-validation.

Figure 1 depicts the confusion matrices of the categorization.

Figure 1 shows that the Non-TB class has the least amount of classification error. This is because it has an accuracy of 98.99% when categorizing the non-TB class. In comparison, the TB classes DS-TB, DR-TB, MDR-TB, and XDR-TB have accuracies of 92.2, 97.1, 88.7, and 91.1 percent, correspondingly. They have an error rate of zero when classifying non-TB and DS-TB; however, they have an error rate of two, one, and one times when the final result is DR-TB, MDR-TB, and XDR-TB, rather than having the Non-TB class. However, classes DR-TB, MDR-TB, and XDR-TB have the largest errors to predict as classes DS-TB, XDR-TB, and DR-TB, accordingly. The highest classification error of the prediction class DS-TB is that it classifies to be class MDR-TB.

Based on the data shown in Table 1, we can determine the contribution of each step added to the proposed model. Table 2 displays the average AUC, F-measure, and Accuracy of the models.

The results obtained in Table 2 indicate that adding image segmentation into the models yields a 26.13 percent better answer than when it is not included. Moreover, data augmentation can improve the quality of a solution by 6.61%, compared to those that do not employ it. In conclusion, the AMIS-WLCE solution is superior to that of UWA by 5.83%, and to that of AMIS-WCE by 2.13%. Therefore, we may infer that the model that uses image segmentation, augmentation, and AMIS-WLCE as the fusion approach is the best proposed method for testing with other methods. The comparative efficacies of the proposed model and the existing procedures are evaluated. As part of this experiment, we will perform the comparison of the optimal approach identified in previous section with the heuristics that are already in use. The comparison will be carried out as a binary classification, and the results will be displayed in Table 3, Table 4, Table 5, Table 6 and Table 7.

Using the data shown in Table 10, we compute the mean difference between our proposed approaches and other methods using Equation (1). The calculation result is displayed in Table 4.
(1)$$Ave\;\left(\%\right)=\frac{F_{m}^{P}F_{m}^{C}}{F_{m}^{C}}\times 100\%$$
where $\;F_{m}^{C}$ is the value of KPI m for the competitor’s approach, and $F_{m}^{P}$ is the value of KPI m for the proposed method. According to Table 10, the average accuracy of the proposed method is 21.7% more than those of the methods previously described in the literature. The AUC and F-measure of the suggested technique are 16.1% and 12.6% greater than those of prior research. The classification of DS against DR yielded the greatest improvement when compared to other categories. The proposed method yields a solution that is 26.9% superior to others, while DS against MDR yields the second most improved solution. A classification between DS and XDR yields the least improvement for the suggested strategy when compared to other classifications. The proposed method deviates from the earlier literature by 16.8 percent, on average. In Table 5, Table 6 and Table 7, a comparison is provided between the proposed method and the previously published research on TB detection using the CXR images of datasets A, B, and C, respectively.

On the basis of the data shown in Table 5, Table 6 and Table 7, it can be concluded that the proposed method delivers better solutions than the previously described method. The suggested method yields, on average, 12.6% better solutions than the earlier literature for dataset A, 23.4% better solutions for dataset B, and 32.8% better solutions than the current state-of-the-art methods published to date for dataset C. A value of 70.8% is the largest difference between the suggested approach and other methods, which is the difference between the method proposed by Sandler et al. [56]. A value of 0.4% is the least difference between the suggested method and the one proposed by Ahmed et al. [8].

Based on the results presented in Tables 10–14, it can be concluded that the suggested model provides a superior solution for categorizing binary data compared to the models offered in the earlier literature. The suggested method’s lowest accuracy is 88.6%, which corresponds to the classification of MDR and XDR, while its highest accuracy is 99.6%, which corresponds to the classification of TB-positive patients in dataset B. In conjunction with the data from previous researches, the maximum accuracy of the proposed method for classifying the multiclass model is 92.6%, which falls within the acceptable range for medical personnel. In the following part, the proposed model will be incorporated into the web application serving as the TB-DRC-DSS. The subsequent part is the outcome of the TB-DRC-DSS

#### 2.1.2. The Web Application Result

The URL for the online application is https://itbru.com/TDRC01/ (accessed on 9 November 2022). Figure 2 is an example of the TB-DRC-DSS that we designed to detect TB and to classify drug-resistant strains. The physicians who utilize TB-DRC-DSS are TB specialists (pulmonologists and internal medicine). The administrator of the TB-DRC-DSS is the pharmacist, who must prescribe the TB-treatment regimen to the physician and inform him or her of the currently recommended regimen.

Following the completion of the TB-DRC-DSS system’s design, two experiments will be conducted with the finalized system. From the aggregated dataset, 428 CXR images were selected at random from the test dataset. These photos will be used as input CXR images for the test that will determine whether or not the TB-DRC-DSS can accurately classify CXR images. The results of the conducted tests are displayed in Table 8. During the second part of the investigation, TB-DRC-DSS testing will commence. A total of 31 pulmonologists and internal medicine specialists in Thailand have been selected to evaluate the TB-DRC-DSS between 1 September 2022 and 30 September 2022. Then, we ask them via an online survey of what they think of the TB-DRC-DSS, and the findings are shown in Table 9 below.

From Table 8, the TB detection has 96.3 accuracy while DS-TB, DR-TB, MDR-TB, and XDR-TB have 91, 91.9, 92.2, and 92.5% accuracy, respectively. The average accuracy of all types of classification is 92.8%.

With scores of 9.42 and 9.45 out of a possible 10, the physicians believed that the TB-DRC-DSS provides a quick and accurate response. They gave the application a score of 9.39 for assisting their diagnostic procedure, 9.48 for reliability, and 9.52 for their preference.

## 3. Discussion

This research aims to introduce an artificial intelligence (AI) model that can diagnose tuberculosis and classify drug resistance using a CXR image. Four parts make up the suggested method: (1) image segmentation, (2) image augmentation, (3) the formation of an ensemble CNN model, and (4) the development of an efficient decision fusion strategy. After obtaining an effective model, it will be incorporated into a web application for usage as a decision support system for TB detection, and as a drug resistance classification-based diagnostic tool.

Regarding the experimental results presented in Section 2, lung segmentation can improve the quality of the solution by 26.13 percent, compared to a solution without lung segmentation. This finding is consistent with the conclusions of Karki et al. [9], Shiraishi et al. [60], Ronneberger et al. [61] that lung segmentation can improve the quality of the solution by removing unrelated elements from the images and making it easier for the AI to identify abnormalities in the lung without noise interfering with the AI’s diagnosis. In addition to these three research findings that image segmentation improves the quality of the TB diagnosis solution, Lui et al. [62] provides evidence that effective lung segmentation can greatly improve the quality of the final prediction. Lui et al. [62] enhance the lung segmentation of U-Net into the “automated segmentation” lung segmentation technique. They improve the classic U-Net network by utilizing Efficientnet-b4 for pre-training. In our research, we employ Efficientnet-b7 instead of Efficientnet-b4 to increase the solution quality of the segmentation, and the results demonstrate that the lung segmentation does actually improve the final prediction quality. Due to its ability to exclude CXR picture elements that are unrelated to TB detection, lung segmentation can enhance the quality of the solution. As a result, the lung can be seen more clearly by the AI, and the AI can better classify the CXR.

Image or data augmentation is one option for enhancing the quality of the predictions of deep learning. Image augmentation is a method of modifying existing images to provide more training data for a model. In other words, it is the process of augmenting the dataset available for training a deep learning model. The computational result indicates that models that use data augmentation enhance the solution quality by 6.61 percent, compared to those that do not. This conclusion is consistent with Momeny et al. [18], Tasci et al. [22], and Shiraishi et al. [60], which state that data augmentation is extremely beneficial for the data preparation of the deep learning model. Shiraishi et al. [60] used chest X-ray (CXR) images provided by COVID-19 patients, and used a combination of classic data augmentation techniques and generative adversarial networks (GANs) in order to screen and classify the patients. By applying a variety of filter banks, including Sobel, Laplacian of Gaussian (LoG), and Gabor filters, it has made it possible to extract characteristics at a deeper level. It demonstrates that an excellent result was achieved in detecting COVID-19 patients using CXR. This demonstrates that an excellent preprocessing strategy, such as data augmentation, was responsible for effective classification across a wide variety of data types and aspects of the study. We can conclude that image segmentation and augmentation are required for deep learning, which is employed to detect TB-positive and drug-resistant types. Consequently, enhancing and utilizing effective picture preprocessing is crucial for the final classification outcome.

Methods employing a single CNN architecture of machine learning and deep learning models are compared with our suggested technique. These include VGG16 [53], ResNet50 [54], EfficientNet [55], MobileNetv2 [56], and FuseNet [57]. In terms of classification accuracy, the suggested technique surpassed these methods. The proposed method can enhance the solution quality by a maximum of 23.4 percent for dataset A, 70.8 percent for dataset B, and 117.5 percent for dataset C. This is because ensemble deep learning has been employed in this study. Ensemble deep learning uses many CNN architectures to create the model, which is in fact superior to using a single CNN design. Tasci et al. [22], Khatibi et al. [24], Badrinarayanan et al. [63], and Jain et al. [64] confirm that ensemble deep learning beat the classic machine learning and deep learning model. The choice of CNN architecture is also important to the quality of the final image classification solution. MobileNetV2 is one of the CNN architectures that has been frequently employed in mobile application research and development. MobileNetV2 is a rapid and effective approach for solving the image classification problem, as evidenced by the fact that Sandler et al. provided the effective model utilizing MobileNetV2 to address a variety of problems. EfficientNetB7 and ResNet50 are larger models than MobileNetV2, and they have been successfully applied to image classification issues [55,56]. In our suggested model, these three CNN architectures have been uniformly incorporated. It has been demonstrated that the heterogeneous ensemble deep learning model is more effective than the heterogeneous deep learning model at classifying drug-resistant models [65], which is consistent with our finding that using the proposed model significantly improves solution quality, as shown in Table 2, Table 3, Table 4, Table 5 and Table 6.

Numerous studies have developed excellent fusion procedures, such as the average weight model, majority voting, and optimum weight via AMIS [16,58], which are necessary for the ensemble deep learning model [16,60]. This study employs UWA, AMIS-WCE, and AMIS-WLCE. The research demonstrated, in accordance with Han et al. [16], Li et al. [58] and Prasitpuriprecha et al. [65], that the AMIS-WLCE improves the solution quality of the existing model.

Figure 3 demonstrates the GradCAM heatmap [66,67,68] of various drug-resistant types. We divide the entire lung into six parts: left and right at the top, center left and right, and bottom left and right. For the classification of non-TB, the CNN will use all six areas, whereas for DS-TB, the system will use the lung’s interior from top to bottom. For DR-TB, MDR-TB, and XDR-TB, the system will use different lung sectors to identify each type of tuberculosis. For instance, for DR-TB, the system will focus on the top left and right sectors. MDR-TB and XRD-TB use comparable regions to detect drug resistance. These regions include the lower left and right sides, and the upper left and right sides of the lung. The distinction between MDR-TB and XDR-TB is that MDR-TB utilizes the inner portion of the lung, whereas XDR-TB uses the outside portion.

A web application serves as a communication tool for developers and medical personnel. The designers of the application hope to enhance the TB detection and drug response classification processes. We determined, based on the computational results, that the medical personnel preferred and had faith in the diagnosis-supporting application. Due to the fact that the web application is user-friendly and can communicate with the user more effectively than the first generation of web applications, the internet provider is also simple to access in the present technological environment. These results are consistent with Jarworski [43], Kokelj et al. [46] Buels et al. [47], Jiménez et al. [48] and Jacinto [49]. They created a user-friendly 2D and 3D web-based application for the medical service, which the user prefers to use, and which can boost the patients’ preference in the current medical service environment.

## 4. Materials and Methods

This research intends to improve the accuracy of classifying TB-positive and TB-negative patients, as well as classifying the drug responses of tuberculosis patients, utilizing ensemble deep learning. Therefore, the CXR image will be utilized to classify the five classes: TB negative (non-TB), DS-TB, DR-TB, MDR-TB, and XDR-TB. After acquiring a suitable algorithm to classify them, the web application will be deployed as a decision support system for medical personnel detecting tuberculosis (TB). The system is known as the TB detection and drug-resistant classification decision support system, or TB-DRC-DSS for short. Figure 4 illustrates the structure of this investigation.

The TB-DRC-DSS development framework is depicted in Figure 4.The TB-DRC-DSS was developed through a process that consisted of the following three steps: (1) collecting the CXR dataset, and the efficacy of previous methods from a variety of literature; (2) developing the artificial multiple intelligence system deep learning algorithm (AMIS-EDL) in order to construct a model that can differentiate TB from non-TB, and various types of drug responses; and (3) developing the TB-DRC-DSS as a web application. The following is a detailed explanation of each level.

### 4.1. Revealed Dataset and Compared Methods

The datasets utilized in this investigation are compatible with those utilized in Shi et al. [69] and Ahmed et al. [8]. It contains a total of four datasets for evaluating the efficacy of the proposed approach and comparing it to the most recent methods proposed in the literature.

Portal Dataset: This dataset is accessible at https://tbportals.niaid.nih.gov under the name Portal dataset [70] (Accessed on 2 August 2022). The initial dataset is one that Karki et al. [9] employed. There are a total of 5019 CXR images linked with tuberculosis in the dataset, of which 1607 are DS-TB and 468, 2087, and 857 are DR-TB, MDR-TB, and XDR-TB images, respectively. It is important to keep in mind that the drug susceptibility label that is associated with each image is not produced from the image itself; rather, it is obtained through drug susceptibility testing.

Dataset A: Dataset A consists of a combination of two datasets. The Montgomery County CXR dataset contains 138 X-rays from the Montgomery County’s TB screening program, whereas the Shenzhen CXR dataset contains 62 images [71].

Dataset B: Dataset B is a labeled dataset that was compiled by a variety of institutes operating under the auspices of the Ministry of Health in the Republic of Belarus [72]. The TB-infected individuals depicted in these 306 images come from 169 distinct people.

Dataset C: The Kaggle public open repository [73] was the source of the acquisition of the labeled dataset known as Dataset C. This dataset contains a total of 6432 images, with 1583 of those images belonging to normal classes, 4274 of those images belonging to pneumonia classes, and 576 of those images belonging to COVID-19 classes.

Each dataset was divided into two subsets: the training dataset (80%) and the test dataset (20%). Table 10 displays the dataset’s details. Datasets A, B, and C will be compared with the results shown in Ahmed et al. [8], Simonyan and Zisserman [53], Wang et al. [54], Tan and Le [55], Sandler et al. [56], Abdar et al. [57], Li et al. [58], and Khan et al. [59]. Table 10 will provide further information regarding the total number of datasets. The portal dataset is compared with the result shown in Ureta and Shrestha [11], Tulo et al. [12], Jaeger et al. [13], Kovalev et al. [14], Tulo et al. [10], and Karki et al. [9].

Based on the information provided in Table 10, we will use datasets A, B, and C, and the Portal dataset to classify tuberculosis and drug-resistant strains into five distinct classes. Table 11 displays the details of the data that will be used in this stage. This dataset includes 406 and 1583 non-TB cases from datasets A and C, as well as 1607,468,2087, and 857 DS-TB, DR-TB, MDR-TB, and XDR-TB cases, respectively. Table 2 displays the result of the combining of the datasets. Framework of the TB/non-TB and Drug-Resistant dataset was shown in Figure 5.

Using three distinct types of key performance indicators (KPIs), the comparative effectiveness of the proposed method to the existing methods described in the literature was evaluated. The area under the curve (AUC), the F-measure, and accuracy are KPIs used to evaluate the efficacy of various approaches in this research. The Receiver Operator Characteristic (ROC) curve is one of the evaluation metrics that can be used for binary classification problems. A classifier’s ability to differentiate between distinct groups is evaluated based on a metric known as the Area Under the Curve (AUC), which also acts as a summary of the ROC curve. When calculating the F-measure, the harmonic mean of recall and precision is employed, and each factor is given equal weight in the calculation. It is helpful for explaining the performance of the model and when comparing models, since it allows for an evaluation of a model while taking into consideration both precision and recall using a single score. It is frequent practice to describe the performance of the model with regard to all classes in terms of its accuracy metric. It is helpful when each class is given the same amount of importance. The formula for calculating it is to divide the total number of guesses by the number of predictions that turned out to be accurate. Table 12 will show the efficiency of the previous research by collecting the AUC, F-measure, and accuracy of the selected previous research [65].

### 4.2. The Development of Effective Methods

The classification model will be developed with the aid of ensemble deep learning. The training and testing datasets will be forwarded to image segmentation so that just the lung region of the CXR picture will be cropped. Later, the training dataset will be sent to data augmentation, while the testing dataset will be used to evaluate the classification model derived from the training dataset. After augmenting, the chosen CNN will be utilized to construct the model. EfficientNetB7, MobileNetV2, and DenseNet121 are the three effective CNN architectures used in this study. Following the formation of the solution model by the CNNs, the decision fusion strategy (DFS) will be employed to produce the final determination. This study will employ three types of DFS. These three models are majority voting, meta-learning, and multiclass weight-AMIS. After obtaining the model using the training dataset, the generated model will be evaluated using the test dataset. Figure 6 illustrates the proposed framework’s fundamental structure. This model is designed to figure out if a patient has tuberculosis, and if they do, what type of drug resistance they have.

After evaluating the classification model and determining that its accuracy is satisfactory, the model will serve as the input model for the web application used to diagnose TB and drug resistance in patients. In the TB-DRC-DSS, the treatment regimen is also recommended to the user, just in case the patient has drug-resistant symptoms. The regimen for TB changes, often according to the disease, and thus the patients must quickly adapt. Therefore, pharmacists must often alter the update of the prescription. The methodology to achieve the framework presented in Figure 6 will be explained stepwise, as follows.

### 4.3. Image Segmentation

In this research, modified U-Net [62] has been applied in the step of image segmentation. U-Net [61] is clearly one of the most effective systems in the field of image segmentation tasks, particularly medical image segmentation. U-Net makes use of skip connection at the same stage, as opposed to depending on direct supervision and loss back transmission on high-level semantic characteristics such as FCN [74], SegNet [63], and Deeplab [75]. This makes it possible for pieces of varied sizes to be fused together, which guarantees that the finally recovered feature map will have more low-level features. As a result, both multi-scale prediction and deep supervision are achievable. Upsampling also increases the precision of the image’s information, such as the edge that was restored after segmentation. One of the limitations of deep learning for medical image processing is that it typically supplies a small number of instances. Despite this limitation, U-Net continues to work fairly effectively. U-Net was employed as the foundation for the architecture of the automatic lung segmentation model due to these benefits. The input dimensions for our experiment with ImageNet’s pre-trained base networks were 256 × 256 × 3, while the output dimensions were 256 × 256 × 1. This specific method employs a network design with five coding and five decoding levels. The encoder has been pre-trained using ImageNet and Efficientnet-b4.

The decoding block is the primary source of novelty in the model employed in this research. The decoder includes five blocks; each decoding layer consists of a dropout layer, a two-dimensional convolution and padding layer, and then two residual blocks and a LeakyReLU. Additionally, we attempt to combine three residual blocks into each decoding block, but the model’s performance is not better. The purpose of the dropout layer is to enhance the model’s ability to generalize and to prevent the model from overfitting. The two-dimensional convolution layer continues to extract the image data. Two residual blocks [76] can prevent the “vanishing gradient” and enhance information dissemination.

Residual block is the most important module in Resnet [77]. It adds a quick connection between the input and output of network layers. In other words, it directly adds the original information and output without any change. The deeper the network is, the more obvious the “vanishing gradient”, and the training effect of the network will not be incredibly good. However, the shallow network now cannot significantly improve the network performance. That is a contradictory problem, but the residual block effectively solves the contradiction of avoiding the “vanishing gradient” when deepening the network. Figure 4 shows the architecture of the modified U-net proposed by Lui et al. [78] when the EfficientNetB7 has been used instead of Efficientnet-b4, similar to the one that is used in Ronneberger et al. [61].

From Figure 7, LeakyReLU [78] was employed to serve as the activation function. LeakyReLU’s function is similar to the ReLU algorithms. The only distinction occurs in the case where the input is greater than 0. The value of the component of the function where the input of ReLU is less than 0 is always 0, whereas the value of the portion where the input of LeakyReLU is less than 0 is always negative and has a very little gradient. The activation function of the middle layer is set to LeakyReLU, and following the application of a 1 × 1 convolution layer, the “Sigmoid” activation function is employed to construct the mask.

### 4.4. Data Augmentation

The objective of data augmentation is to increase the volume and variety of the training datasets. This extension is carried out by generating synthetic datasets. The augmented data can be viewed as having originated from a distribution that closely resembles the actual distribution. After then, the expanded dataset will be able to explain more complete qualities. The techniques for augmenting image data can be used for a variety of data types, including object recognition [79], semantic segmentation [80], and image categorization [81].

Basic image augmentation techniques include image manipulation, erasure, image mixing, and image transformations. The image transformation techniques include rotating, mirror, and crop. The majority of these techniques directly alter photos and are straightforward to execute. However, disadvantages exist. First, original image alterations only make sense if the existing data closely match the distribution of the actual data. Secondly, some fundamental picture alteration techniques, such as translation and rotation, are susceptible to the padding effect. In other words, after the operation, some of the images will be pushed beyond the boundary and lost. Therefore, a variety of interpolation techniques will be utilized to fill in the missing data. Typically, the region outside the image’s border is assumed to have a constant value of 0 and will be dark after modification. Figure 8 displays an example of image enhancement using multiple enhancement techniques, as explained previously.

### 4.5. CNN Architectures

We will utilize the heterogeneous CNN ensemble for our deep learning ensemble in this study. In our scenario, three distinct CNN architectures will be implemented. These are MobileNetV2 [56], EfficientNetB7 [55], and DenseNet121 [74,82]. MobileNetV2 was initially proposed by Sandler et al. [56], and it is a lightweight network that reduces the weighted parameters by employing depth-wise separable convolutional (DwConv) layers and the inverted residuals of the bottleneck block. Tan and Le [55] were the ones who initially presented EfficientNetB7 with the intention of searching the hyperparameters of CNN architectures. These hyperparameters include width scaling, depth scaling, resolution scaling, and compound scaling. In order to obtain an informative channel feature using GAP summation, squeeze-and-excitation (SE) optimization was also introduced to the bottleneck block of EfficientNet. The fundamental concept of Densely Connected Convolutional Networks was utilized in DenseNet121, which was based on the work of [76]. The connectivity pattern between layers that is introduced in previous architectures is made more straightforward by DenseNets. Highway Networks, Residual Networks, and Fractal Networks are the names given to these interconnected designs. All of the architectures that have been described up to this point will be implemented in the ensemble deep learning system in a heterogeneous fashion, as illustrated in Figure 9.

### 4.6. Decision Fusion Strategy

The decision fusion strategy (DFS) integrates the classification judgments of many classifiers into a single related conclusion. Multiple classifiers are typically applied with multi-modal CNN when it is difficult to aggregate all of the CNN’s outputs. In this study, three distinct decision fusion strategies will be implemented. These strategies are the unweighted average model (UAM), the AMIS-weighted classifier ensemble (AMIS-WCE) [64,83], and the AMIS-weight label classifier ensemble (AMIS-WLCE) [84,85].

To perform all three fusion procedures, UAM will use Equation (2) while AMIS-WCE and AMIS-WLCE will use Equations (3) and (4) to determine the prediction value for classifying TB and drug-resistant types. $Y_{ij}$ is defined as the value to predict CNN *i* class *j* before the operation of Equations (2)–(4). $V_{j}$ is the value which is used to classify class *j* after the fusion of different CNN results. $W_{i}$ is weight of CNN *i*, and $S_{ij}$ is the weight of CNN *i* to classify class *j*. *i* is the number of CNNs used in the ensemble model.
(2)$$V_{j}=\frac{\sum_{i=1}^{I}Y_{ij}}{I}$$
(3)$$V_{j}=\overset{I}{\underset{i=1}{\sum }}W_{i}Y_{ij}$$
(4)$$V_{j}=\overset{I}{\underset{i=1}{\sum }}S_{ij}Y_{ij}$$

The final prediction will be executed by the classify class *j* that has a maximum value of $V_{j}.\;$AMIS will be used to find the optimal values of $W_{i}$ and $S_{ij}$. AMIS is the first proposed metaheuristic in [84]. AMIS has five steps: (1) generate an initial work package (WP) set; (2) pick the intelligence box or the improvement box (IB); (3) all WPs execute the selected WP; (4) update the heuristics information; and (5) iteratively complete stage 2 to step 4 until the termination condition is fulfilled. Typically, the number of iterations serves as the halting condition. The dimension of the WP will be *D*, and *NP* is the number of *WP* that will be run at each iteration.

Each WP will independently choose an IB to improve the current solution. The selection of which WP to use has a distinct probability function, as seen by Equation (5). In each repetition, the WP will select the IB based on the outcome of a roulette wheel selection method.
(5)$$P_{bt}=\frac{FN_{bt-1}+\left(1-F\right)A_{bt-1}+KU_{bt-1}}{\sum_{b=1}^{B}FN_{bt-1}+\left(1-F\right)A_{bt-1}+KU_{bt-1}}$$

The probability of IB *b* in iteration *t* is denoted by $P_{bt}$. *F* is the scaling factor, which is set to 0.7 as the recommendation from Pitakaso et al. [77]. $A_{bt-1}$ is the average objective function of all WPs that chose IB *b* from iteration 1 to iteration *t* − 1. *K* is a predetermined integer value set to 3 [77]. $U_{bt-1}$ is the positive integer that will rise by 1 if the iteration’s best solution is in IB *b*. $N_{bt-1}$ is the total number of WP that selected IB *b* from iteration 1 to iteration *t* − 1, each of which has to be iteratively updated. The details of the updating are shown in Table 13.

In this study, eight intelligence boxes will be used, and they are shown in Table 14 (Equations (6)–(14)).

The design of the IB equation in accordance with Gardner’s idea is presented in his book [76]. $X_{elt}$ is the value of WP *e* element *l* in iteration *t,* while *r*, *n*, and *m* are non-equal elements of the set of WP (1 to E) not equal to e. $B_{l}^{gbest}$ is the global best WP. $H_{el}$ is a random number of WP *E* element *l*. F1 and F2 are the scaling factors, which are defined as 3 (as suggested by [77]), and CR is the crossover rate, which [77] recommends is equal to 0.8. $R_{elt\;}$ is a WP generated at random from WP *e* element *l* at iteration *t*. $B_{el}^{pbest}$ *t* is the optimal solution discovered by WP *e*. Equation (15) is used to update the value of $X_{elt+1}$, and $X_{elt+1}$ is related to the value of $W_{i}$ and $S_{ij}$, as shown in Equations (16) and (17).
(15)$$X_{elt+1}=\{\begin{array}{c} Z_{elt}\;\;\;\;\;if\;\;f\left(Z_{elt}\right)\leq \;f\left(X_{elt}\right)\;\;and\;update\;f\left(X_{elt}\right)=f\left(Z_{elt}\right)\\ X_{elt+1}\;\;otherwise\;\;\;\;\;\;\;\;\;\;\;\;\;\;\;\;\;\;\;\;\;\;\;\;\;\;\;\;\;\;\;\;\;\;\;\;\;\;\;\;\;\; \end{array}$$
(16)$$W_{i}=X_{eit}\;\;\;\mathrm{for all}\;e\;\mathrm{and}\;t$$
(17)$$S_{ij}=X_{e\left(i+j\right)t}\;\;\;\mathrm{for all}\;e\;\mathrm{and}\;t$$

Examples of using different fusion strategies to search for the classification result are shown in Table 15.

Table 15 demonstrates that if the unweighted average model is utilized, class C-2 will be picked; however, classes C-1 and C-3 will be selected if the AMIS-WCE and AMIS-WLCE fusion strategies are utilized. The result shown in Table 14 is the outcome of utilizing WP one to decode for the classification result. All WPs must perform in the same manner in order to identify the WP that provides the best answer.

In this article, we explore image segmentation (Seg), augmentation (Aug), and ensemble deep learning using a heterogeneous ensemble architecture, and three decision fusion techniques. These techniques are UWA, AMIS-WCE, and AMIS-WLCE. Consequently, the combination of the aforementioned strategies can produce solutions of varying quality. We use the DOE to design the experiment, and Table 16 outlines the experimental design of the proposed approaches for classifying TB against non-TB and drug resistance level.

### 4.7. TB-DRC-DSS Design

Internet-based software applications are known as web apps. Contrary to traditional computer programs, which are installed and run locally on the user’s computer, they are hosted on a separate server. A computer with an internet browser and an active internet connection is all that is required to access them. The goal of good web design is to provide the best possible user experience by enhancing both the user interface (UI) and the user experience (UX) of a website or web application. Different from software development, the web design is concerned with the website’s usability, accessibility, and aesthetics. Figure 10 is the web application architecture design that we used to develop TB-DRC-DSS.

From Figure 10, web applications were developed using HTML, JavaScript, and TensorFlow, under the responsive design concept, making them available on a wide range of devices including computers, mobile phones, or tablet computers. Users (medical staff) can upload image files that they want to analyze with their personal computers. This image file is forwarded to a server to diagnose the disease and to suggest the appropriate drugs and medications for the predicted diseases. The treatment regimen is updated by the pharmacist who takes responsibility for this issue. Once the system obtains the image, the deep learning model is applied to predict the type of disease. Once the type of disease is obtained, it will search for detailed information and recommendations for the medication used in that particular disease, and send it back to the user’s side.

## 5. Conclusions 

This study proposes a heterogeneous ensemble deep learning model to detect multiple classes of TB-positive and drug-resistant strains. The built model has been incorporated into the online application to aid in the TB diagnosis process, and to recommend a treatment protocol for TB-positive patients. As image preparation approaches for the deep learning model, image segmentation and augmentation have been implemented. Three distinct CNN architectures were employed to build the classification model. These are MobileNetV2, DenseNet121, and EfficientNetB7 designs. As a manner of ensemble, the heterogeneous ensemble model has been employed. Three decision fusion strategies are employed. UWA, UWA, AMIS-WCE, and AMIS-WLCE are these three methodologies. The aggregate of the Portal dataset [70], dataset A [71], and dataset C [73] is the dataset utilized to validate the developed model. The total number of images in the test and train datasets is 7008 images. The binary and multiclass categorizations of these datasets will be compared to the categories found in Ahmed et al. [8], Karki et al. [9], Tulo et al. [10], Ureta and Shrestha [11], Tulo et al. [12], Jaeger et al. [13], Kovalev [14], Simonyan and Zisserman [53], Wang et al. [54], Tan and Le [55], Sandler et al. [56], Abdar et al. [57], Li et al. [58] and Khan et al. [59]. 

The computational result can be summed up in three parts: (1) a comparison between the provided models; (2) a comparison with the current approaches from the prior literature; and (3) an evaluation of the web application. The most effective method for classifying multiclass TB-detection and drug-resistant types is a model composed of (1) image segmentation; (2) data augmentation; and (3) the AMIS-WLCE fusion strategy. The optimal model described above has an average accuracy that is 19.74% higher than other existing approaches, with maximum and minimum accuracies that are 43.57 and 1.65% higher, respectively. Image segmentation can increase the quality of the solution by 26.13 percent, while data augmentation and AMIS-WLCE can improve it by 6.61 percent and 2.13 percent, respectively.

The results obtained in Table 8 indicate that adding image segmentation into the models yields a 26.13 percent better answer than when it is not included. Moreover, data augmentation can improve the quality of a solution by 6.61% compared to those that do not employ it. In conclusion, the AMIS-WLCE solution is superior to that of UWA by 5.83%, and to that of AMIS-WCE by 2.13%. Therefore, we may conclude that the model that employs image segmentation, augmentation, and AMIS-WLCE as the fusion method is the best proposed way for comparing with other approaches. For multiple classes of TB-positive and drug-resistant strains, the suggested model’s accuracy is 92.6%. The suggested model is utilized to analyze the binary classification of tuberculosis and non-tuberculosis, as well as DS-TB against DR-TB and other pairwise comparisons between each type of drug-resistant categorization. The computational result demonstrated that the classification accuracy of DS-TB against DR-TB has an averaged 43.3% improvement over other approaches. The categorizations between DS-TB and MDR-TB, DS-TB and XDR-TB, and MDR-TB and XDR-TB had an average accuracy improvement of 28.1%, 6.2%, and 9.4% over other approaches. Therefore, we can infer that the proposed model improves the solution quality of the existing model used to classify the binary classification model, and it is the first model capable of classifying TB/non-TB patients and the type of treatment resistance in positive TB patients.

The generated model was incorporated into the TB-DRC-DSS, and the effectiveness of the TB-DRC-DSS was evaluated using two approaches. The first experiment conducted to evaluate the accuracy of the TB-DRC-DSS revealed that it can categorize with 92.8% accuracy all 428 randomly selected datasets. The second experiment is the trial phase of using TB-DRC-DSS in the real world, and we discovered that 31 medical staff gave the TB-DRC-DSS a preference score of 9.52 out of a possible 10 points, 9.42 for believing that the TB-DRC-DSS result is the correct answer, and 9.32 and 9.39 for believing that the TB-DRC-DSS will make their work easier.

Regarding our research, there are three research gaps that must be filled in the future. First, the accuracy of the classification of multiclass drug-resistant types and the detection of tuberculosis (TB) is not yet 100 percent; therefore, it would be a fascinating area of research to develop an exceptional model that can make more accurate predictions. The model can incorporate the most efficient image segmentation, augmentation, and normalization. Denoising methods are an intriguing process that can enhance the image segmentation solution quality. Second, the web application should be able to aid in the treatment of TB patients. The application must include information regarding therapy follow-up. Whenever necessary, additional decisions must be taken based on the results of implementing the suggested protocol. Due to the rapid emergence of treatment resistance, the recommended regimen for tuberculosis patients has changed regularly. The pharmacist must adhere to the WHO treatment regimen in order to modify the patient’s treatment regimen, as well as recoding information on the patient’s drug reaction and using data mining to analyze the patient’s data in order to modify the regimen as needed.

## Figures and Tables

**Figure 1 pharmaceuticals-16-00013-f001:**
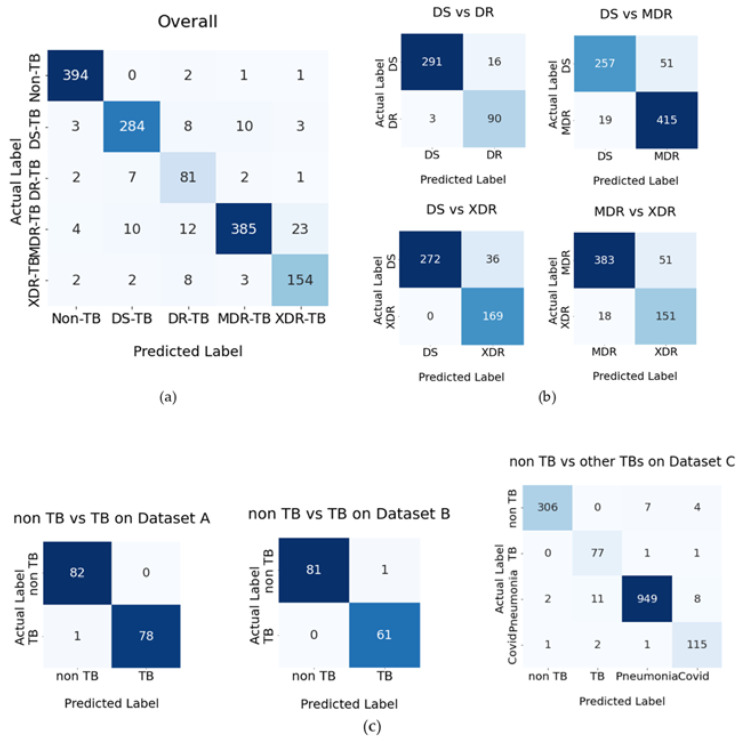
Confusion matrices of the classification. (**a**) Overall result of five classifications, (**b**) the results of the comparison of drug resistance, and (**c**) the results of comparison for non-TB vs. other classifications on datasets A, B, and C.

**Figure 2 pharmaceuticals-16-00013-f002:**
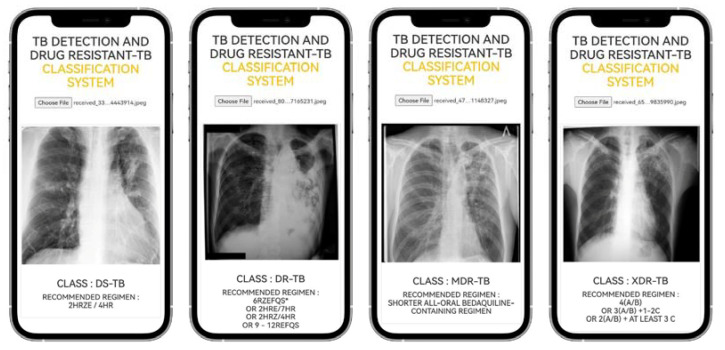
Example of TB-DRC-DSS User Interface.

**Figure 3 pharmaceuticals-16-00013-f003:**
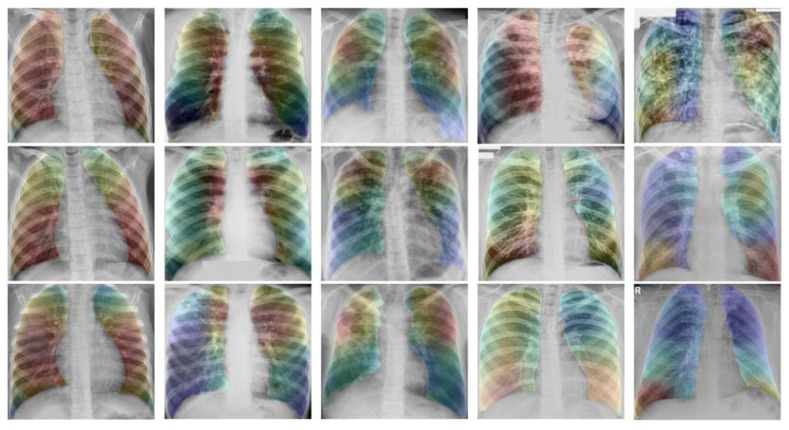
GradCAM visualizations on X-ray images.

**Figure 4 pharmaceuticals-16-00013-f004:**
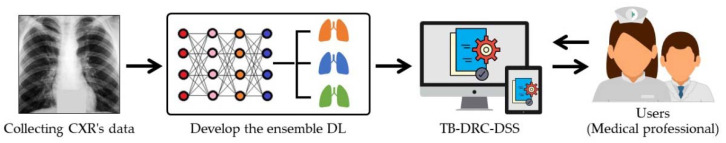
The development of the TB diagnosis supporting system (TB-DRC-DSS).

**Figure 5 pharmaceuticals-16-00013-f005:**
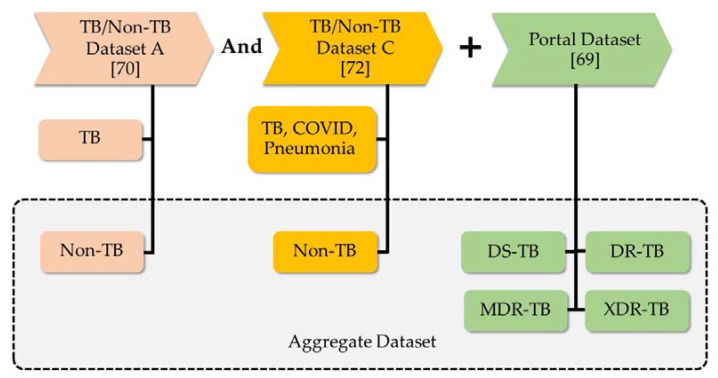
Framework of the TB/non-TB and Drug-Resistant dataset.

**Figure 6 pharmaceuticals-16-00013-f006:**
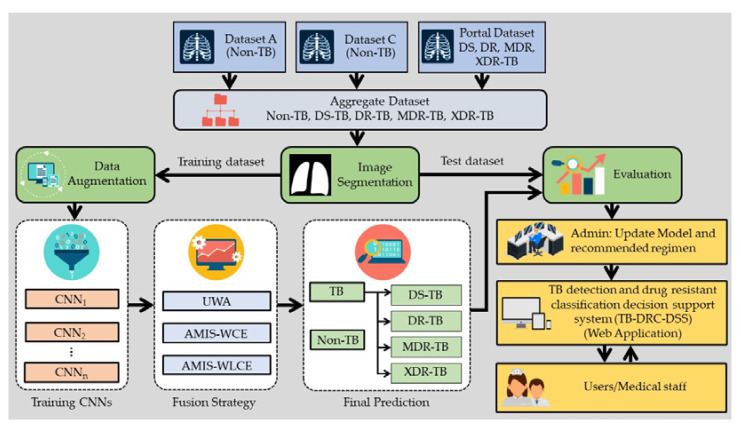
Framework of the proposed model.

**Figure 7 pharmaceuticals-16-00013-f007:**
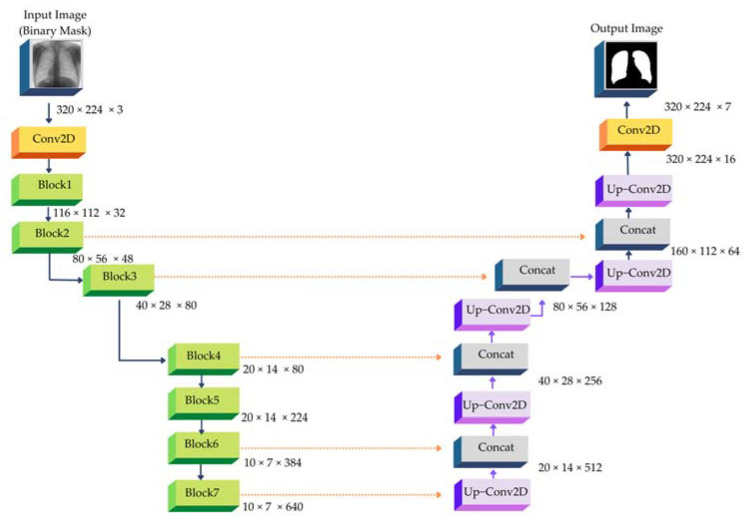
The architecture of U-Net with EfficientNetB7 Encoder (adapted from [63]).

**Figure 8 pharmaceuticals-16-00013-f008:**
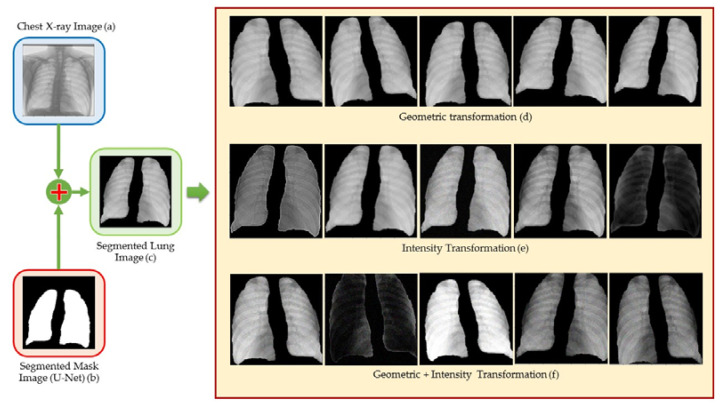
Image augmentation example.

**Figure 9 pharmaceuticals-16-00013-f009:**
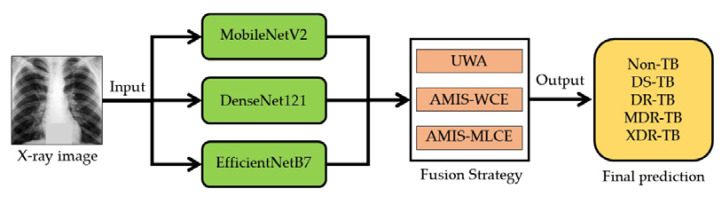
Structure of the EDL using Heterogeneous and Heterogeneous manners.

**Figure 10 pharmaceuticals-16-00013-f010:**
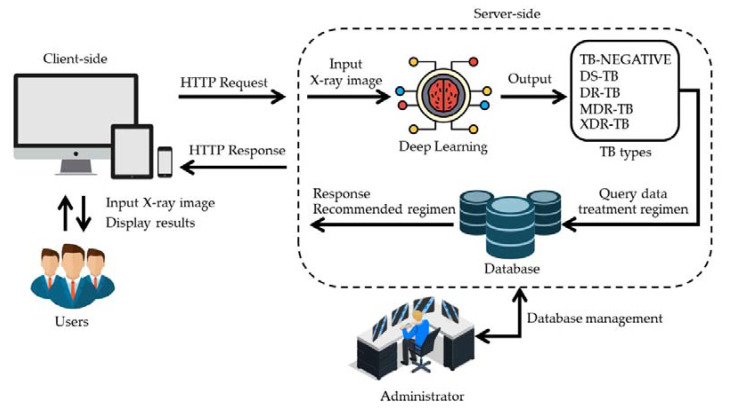
The design framework of web application for the case study.

**Table 1 pharmaceuticals-16-00013-t001:** Computational results of the proposed models.

Methods		5-cv			Test Set	
	AUC	F-Measure	Accuracy	AUC	F-Measure	Accuracy
N-1	69.2 ± 0.29	61.9 ± 0.17	65.3 ± 0.15	68.5	61.3	64.5
N-2	72.8 ± 0.21	66.2 ± 0.27	68.7 ± 0.07	72.4	65.1	67.9
N-3	72.9 ± 0.07	67.8 ± 0.25	70.2 ± 0.18	72.5	66.3	69.4
N-4	76.1 ± 0.17	67.4 ± 0.14	72.4 ± 0.13	75.4	68.9	71.8
N-5	77.6 ± 0.19	72.8 ± 0.09	73.9 ± 0.21	76.3	71.7	73.6
N-6	79.8 ± 0.25	74.2 ± 0.27	76.7 ± 0.09	79.1	73.5	75.4
N-7	88.2 ± 0.29	82.6 ± 0.25	85.3 ± 0.26	87.8	81.4	84.1
N-8	90.3 ± 0.04	84.8 ± 0.18	87.9 ± 0.17	89.2	85.5	87.3
N-9	91.9 ± 0.07	87.5 ± 0.08	89.8 ± 0.05	91.6	85.8	89.3
N-10	91.5 ± 0.24	87.8 ± 0.09	90.1 ± 0.19	90.7	86.1	88.5
N-11	94.8 ± 0.18	91.2 ± 0.28	91.5 ± 0.24	94.1	90.5	91.1
N-12	95.3 ± 0.07	91.7 ± 0.10	93.9 ± 0.08	94.7	90.9	92.6

**Table 2 pharmaceuticals-16-00013-t002:** Average AUC, F-Measure, and Accuracy of the Proposed Models.

KPI	Image Segmentation		Data Augmentation		Decision Fusion Strategy		
	No Image Segmentation	Use Image Segmentation	No Data Augmentation	Use Data Augmentation	UWA	AMIS-WCE	AMIS-WLCE
AUC	74.0	91.4	80.3	85.1	80.6	83.0	84.5
F-measure	72.6	90.2	78.9	83.9	79.2	81.7	83.4
Accuracy	70.4	88.8	77.1	82.2	77.2	80.0	81.7

**Table 3 pharmaceuticals-16-00013-t003:** The accuracy of the proposed methods with the existing heuristics to classify DS-TB, DR-TB, MDR-TB, and XDR-TB.

Methods	Classes	Features	Region in CXR	Accuracy		
				AUC	F-Measure	Accuracy
Ureta and Shrestha [11]	DS vs. DR	CNN	Whole	67.0	-	-
Tulo et al. [12]	DS vs. DR	Shape	Mediastinum + Lung		93.6	-
Kovalev et al. [14]	DS vs. DR	Texture and Shape	Lung	-	-	61.7
Karki et al. [9]	DS vs. DR	CNN	Lung excluded	79.0	-	72.0
**Proposed method**	**DS vs. DR**	**Ensemble CNN**	**Whole**	**97.2**	**94.9**	**95.2**
Jaeger et al. [13]	DS vs. MDR	Texture, Shape and Edge	Lung	66	61	62
Tulo et al. [10]	DS vs. MDR	Shape	Mediastinum + Lungs	87.3	82.4	82.5
**Proposed method**	**DS vs.** **MDR**	**Ensemble CNN**	**Whole**	**92.5**	**90.1**	**90.6**
Tulo et al. [10]	DS vs. XDR	Shape	Mediastinum + Lungs	93.5	87.0	87.0
**Proposed method**	**DS vs. XDR**	**Ensemble CNN**	**Whole**	**95.7**	**92.1**	**92.5**
Tulo et al. [10]	MDR vs. XDR	Shape	Mediastinum + Lungs	86.6	81.0	81.0
**Proposed method**	**MDR vs. XDR**	**Ensemble CNN**	**Whole**	**90.8**	**86.6**	**88.6**

**Table 4 pharmaceuticals-16-00013-t004:** Average difference between the proposed method and the previous methods mentioned in the literature (%).

Pairwise Comparison	AUC	F-Measure	Accuracy
DS vs. DR	34.1	3.3	43.3
DS vs. MDR	23.1	30.2	28.1
DS vs. XDR	2.4	6.4	6.2
MDR vs. XDR	4.8	10.5	9.4
**Average**	**16.1**	**12.6**	**21.7**

**Table 5 pharmaceuticals-16-00013-t005:** The accuracy of the proposed methods with the existing heuristics to classify TB and Non-TB (Dataset A).

	Classes	Features	Region in CXR	Accuracy
Simonyan and Zisserman [53]	TB vs. non-TB	VGG16	Whole	83.8
Wang et al. [54]	TB vs. non-TB	ResNet50	Whole	82.2
Tan and Le [55]	TB vs. non-TB	EfficientNet	Whole	83.3
Sandler et al. [56]	TB vs. non-TB	MobileNetv2	Whole	80.2
Abdar et al. [57]	TB vs. non-TB	FuseNet	Whole	91.6
Li et al. [58]	TB vs. non-TB	MAG-SD	Whole	96.1
Khan et al. [59]	TB vs. non-TB	CoroNet	Whole	93.6
Ahmed et al. [8]	TB vs. non-TB	TBXNet	Whole	98.9
**Proposed method**	TB vs. non-TB	**Ensemble CNN**	**Whole**	**99.4**

**Table 6 pharmaceuticals-16-00013-t006:** The accuracy of the proposed methods with the existing heuristics to classify TB and Non-TB (Dataset B).

	Classes	Features	Region in CXR	Accuracy
Simonyan and Zisserman [53]	TB vs. non-TB	VGG16	Whole	73.3
Wang et al. [54]	TB vs. non-TB	ResNet50	Whole	65.8
Tan and Le [55]	TB vs. non-TB	EfficientNet	Whole	87.5
Sandler et al. [56]	TB vs. non-TB	MobileNetv2	Whole	58.3
Abdar et al. [57]	TB vs. non-TB	FuseNet	Whole	94.6
Li et al. [58]	TB vs. non-TB	MAG-SD	Whole	96.1
Khan et al. [59]	TB vs. non-TB	CoroNet	Whole	93.7
Ahmed et al. [8]	TB vs. non-TB	TBXNet	Whole	99.2
**Proposed method**	TB vs. non-TB	**Ensemble CNN**	**Whole**	**99.3**

**Table 7 pharmaceuticals-16-00013-t007:** The accuracy of the proposed methods with the existing heuristics to classify dataset C.

	Classes	Features	Region in CXR	Accuracy
Simonyan and Zisserman [53]	TB vs. non-TB vs. COVID-19 vs. Pneumonia	VGG16	Whole	64.4
Wang et al. [54]	TB vs. non-TB vs. COVID-19 vs. Pneumonia	ResNet50	Whole	62.7
Tan and Le [55]	TB vs. non-TB vs. COVID-19 vs. Pneumonia	EfficientNet	Whole	45.4
Sandler et al. [56]	TB vs. non-TB vs. COVID-19 vs. Pneumonia	MobileNetv2	Whole	81.7
Abdar et al. [57]	TB vs. non-TB vs. COVID-19 vs. Pneumonia	FuseNet	Whole	90.6
Li et al. [58]	TB vs. non-TB vs. COVID-19 vs. Pneumonia	MAG-SD	Whole	94.1
Khan et al. [59]	TB vs. non-TB vs. COVID-19 vs. Pneumonia	CoroNet	Whole	89.6
Ahmed et al. [8]	TB vs. non-TB vs. COVID-19 vs. Pneumonia	TBXNet	Whole	95.1
**Proposed method**	TB vs. non-TB vs. COVID-19 vs. Pneumonia	**Ensemble CNN**	**Whole**	**97.4**

**Table 8 pharmaceuticals-16-00013-t008:** TB-DRC-DSS accuracy result.

	Number of Input Images	Number of Correct Classifications	%Correct Classification	Number of Wrong Classification	%Wrong Classification
Non-TB	81	78	96.3	3	3.7
DS-TB	78	71	91.0	7	9.0
DR-TB	74	68	91.9	6	8.1
MDR-TB	102	94	92.2	8	7.8
XDR-TB	93	86	92.5	7	7.5
Total	428	397	463.8	31	36.2
Average	86	79	92.8	6	7.2

**Table 9 pharmaceuticals-16-00013-t009:** The questionnaire results from 30 doctors who were trying to use DRCS.

Questionnaire to Pictures	Level of Agreement (Strongly Agree Level is 10)
TB/DR-TDDS provides quick results.	9.42
TB/DR-TDDS provides the correct results.	9.45
TB-DRC-DSS is beneficial to your work.	9.32
You will choose the TB/DR-TDDS to assist in your diagnosis.	9.39
What rating will you give the TB/DR-TDDS about your reliability?	9.48
How would you rate the TB/DR-TDDS in relation to your preferences?	9.52

**Table 10 pharmaceuticals-16-00013-t010:** Dataset information used to compare with the state-of-the-art methods.

	Dataset A [71]		Dataset B [72]		Dataset C [73]				Portal Dataset [70]			
	Non-TB	TB	Non-TB	TB	Non-TB	TB	Pneumonia	COVID	DS-TB	DR-TB	MDR-TB	XDR-TB
Train dataset	324	315	324	245	1266	315	3879	460	1299	375	1653	688
Test dataset	82	79	82	61	317	79	970	116	308	93	434	169
Total dataset	406	394	406	306	1583	394	4,849	576	1607	468	2087	857

**Table 11 pharmaceuticals-16-00013-t011:** Details of the aggregate dataset.

	Aggregate Dataset				
	Non-TB	DS-TB	DR-TB	MDR-TB	XDR-TB
Train dataset	1591	1299	375	1653	688
Test dataset	398	308	93	434	169
Total dataset	1989	1607	468	2087	857

**Table 12 pharmaceuticals-16-00013-t012:** Details of the compared method in existing research.

Studies	Classes	Features	Region in CXR	AUC	Accuracy F-Measure	Accuracy
Ureta and Shrestha [11]	DS vs. DR	CNN	Whole	67.0	-	-
Tulo et al. [12]	DS vs. DR	Shape	Mediastinum + Lung		93.6	
Jaeger et al. [13]	DS vs. MDR	Texture, Shape, and Edge	Lung	66	61	62
Kovalev et al. [14]	DS vs. DR	Texture and Shape	Lung	-	-	61.7
Tulo et al. [10]	DS vs. MDR	Shape	Mediastinum + Lungs	87.3	82.4	82.5
Tulo et al. [10]	MDR vs. XDR	Shape	Mediastinum + Lungs	86.6	81.0	81.0
Tulo et al. [10]	DS vs. XDR	Shape	Mediastinum + Lungs	93.5	87.0	87.0
Karki et al. [9]	DS vs. DR	CNN	Lung excluded	79.0	-	72.0
Simonyan and Zisserman [53]	TB vs. non-TB	VGG16	Lungs	-	-	83.8 (A), 73.3(B),
Wang et al. [54]	TB vs. non-TB	ResNet50	Lungs	-	-	82.2 (A), 65.8(B), 64.4(C)
Tan and Le [55]	TB vs. non-TB	EfficientNet	Lungs	-	-	83.3 (A), 87.5 (B), 62.7 (C)
Sandler et al. [56]	TB vs. non-TB	MobileNetv2	Lungs	-	-	80.2 (A), 58.3 (B), 45.4 (C)
Abdar et al. [57]	TB vs. non-TB	FuseNet	Lungs	-	-	91.6 (A), 94.6 (B), 81.7 (C)
Li et al. [58]	TB vs. non-TB	MAG-SD	Lungs	-	-	96.1 (A), 96.1 (B), 90.6 (C)
Khan et al. [59]	TB vs. non-TB	CoroNet	Lungs	-	-	93.6 (A), 93.7 (B), 94.1 (C)
Ahmed et al. [8]	TB vs. non-TB	TBXNet	Lungs	-	-	98.98 (A), 99.2 (B), 95.1 (C)
**Proposed Method (set A)**	TB vs. non-TB	Ensemble CNN	Lungs	-	-	-
**Proposed Method (set B)**	TB vs. non-TB	Ensemble CNN	Lungs	-	-	-
**Proposed Method (set C)**	TB vs. non-TB vs. COVID-19 vs. Pneumonia	Ensemble CNN	Lungs	-	-	-
**Proposed method**	DR vs. DS	Ensemble CNN	Lungs	-	-	-
**Proposed method**	DS vs. MDR	Ensemble CNN	Lungs	-	-	-
**Proposed method**	DS vs. XDR	Ensemble CNN	Lungs	-	-	-
**Proposed method**	Non-TB vs. DR vs. VS vs. MDR vs. XDR	Ensemble CNN	Lungs	-	-	-

**Table 13 pharmaceuticals-16-00013-t013:** Update method of heuristics information.

Variables	Update Method	
$N_{bt}$	Sum of number of work packages (WP) that have selected IB *b* from iteration 1 through iteration *t*	
$A_{bt}$	$\left(\frac{\sum_{e=1}^{N_{bt}}f_{et}}{N_{bt}}\right)$$\;\mathrm{average}\;\mathrm{objective}\;\mathrm{value}\;\mathrm{selection}\;\mathrm{when}\;f_{et}$ is objective value of WP *e* in iteration *t*.	
$U_{bt}$	$U_{bt}=U_{bt-1}+G$	(6)
	When $G=\{\begin{array}{l} 1\;\mathrm{if and only if black box b has the global best solution at iteration}\;t\\ 0\;\mathrm{otherwise}\;\;\;\;\;\;\;\;\;\;\;\;\;\;\;\;\;\;\;\;\;\;\;\;\;\;\;\;\;\;\;\;\;\;\;\;\;\;\;\;\;\;\;\;\;\;\;\;\;\;\;\;\;\;\;\;\;\;\;\;\;\;\;\;\;\;\;\;\;\;\;\;\;\;\;\;\;\;\;\;\;\;\;\;\;\;\;\;\;\;\;\;\;\;\;\;\;\;\;\;\;\;\;\;\;\;\; \end{array}$	
$B_{l}^{gbest}$	Current global best WP is updated	
$B_{el}^{pbest}$	Current IB’s best WP is updated	
$R_{elt\;}$	Random number is iteratively updated	

**Table 14 pharmaceuticals-16-00013-t014:** IBs used in this research.

Name of IB	IB-Equation	Equation Number
IB-1	$Z_{elt}=\rho X_{rlt}+F1\left(B_{l}^{gbest}-X_{rlt}\right)+F2\left(X_{mlt}-X_{rlt}\right)$	(7)
IB-2	$Z_{elt}=X_{rlt}+F1\left(B_{l}^{gbest}-X_{rlt}\right)+F2\left(B_{el}^{pbest}-X_{rlt}\right)$	(8)
IB-3	$Z_{elt}=X_{rlt}+F1\left(X_{mlt}-X_{nlt}\right)$	(9)
IB-4	$Z_{elt}=X_{rlt}+R_{el}\left(X_{rlt}-X_{nlt}\right)$	(10)
IB-5	$Y_{elt}=H_{el}$	(11)
IB-6	$Z_{elt}=\{\begin{array}{c} X_{elt}\;\;\;\;\;if\;\;H_{el}\leq CR\;\\ R_{elt\;}\;\;\;\;\;\;\;otherwise\;\;\;\; \end{array}$	(12)
IB-7	$Z_{elt}=\{\begin{array}{c} X_{elt}\;\;\;\;\;if\;\;H_{el}\leq CR\;\\ X_{nlt\;\;}\;\;\;\;\;\;\;otherwise\;\;\;\; \end{array}$	(13)
IB-8	$Z_{elt}=\{\begin{array}{c} X_{elt}\;\;\;\;\;\;\;\;\;\;if\;\;H_{el}\leq CR\\ H_{el}X_{elt}\;\;\;\;\;otherwise\;\;\;\; \end{array}$	(14)

**Table 15 pharmaceuticals-16-00013-t015:** Examples of using different fusion strategies to find the classification result of WP 1.

Details of *W* and *Y* Values					UAM		AMIS-WCE		AMIS-WLCE	
CNN (*i*)	Weighted (*Wi*)	Classes (*j*)	Weighted (*W_ij_*)	*Y_ij_*	$\overset{3}{\underset{i=1}{\sum }}\frac{Y_{ij}}{3}$	Prediction Class	$\overset{3}{\underset{i=1}{\sum }}W_{i}Y_{ij}$	Prediction Class	$\overset{3}{\underset{i=1}{\sum }}W_{ij}Y_{ij}$	Prediction Class
M-1	0.5	C-1	0.10	0.6	C-1 = 0.90 C-2 = 0.37 C-3 = 0.33	C-2	C-1 = 0.37 C-2 = 0.34 C-3 = 0.28	C-1	C-1 = 0.12 C-2 = 0.06 C-3 = 0.13	C-3
		C-2	0.20	0.3						
		C-3	0.20	0.1						
M-2	0.2	C-1	0.02	0.1						
		C-2	0.03	0.5						
		C-3	0.15	0.4						
M-3	0.3	C-1	0.03	0.2						
		C-2	0.07	0.3						
		C-3	0.20	0.5						

**Table 16 pharmaceuticals-16-00013-t016:** Combination of different methods used in ensemble deep learning.

#No	No. Seg	Seg.	No. Aug	Aug.	UWA	AMIS-WCE	AMIS-WLCE
N-1	√		√		√		
N-2	√		√			√	
N-3	√		√				√
N-4	√			√	√		
N-5	√			√		√	
N-6	√			√			√
N-7		√	√		√		
N-8		√	√			√	
N-9		√	√				√
N-10		√		√	√		
N-11		√		√		√	
N-12		√		√			√

## Data Availability

Data is contained within the article.

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
