# Peer review of "Drug-Resistant Tuberculosis Treatment Recommendation, and Multi-Class Tuberculosis Detection and Classification Using Ensemble Deep Learning-Based System"

_pharmaceuticals, 2022, doi:10.3390/ph16010013_

Round 1
Reviewer 1 Report
This study presents machine learning models for TB related classification. It's clearly written and explained, but I have a few questions.
1. It does not make any sense to compute the average performance improvement for different metrics, Table 10. I would remove that column.
2. The source code is not listed, which is problematic as other research groups cannot reproduce the results.
3. The TB-DRC-DSS is not where to find and having a URL would be helpful.
Author Response
Thank you for your valuable comments. We'd tried to answer all of your recommendations as below.
|
Comment |
Answer:
|
|
It does not make any sense to compute the average performance improvement for different metrics, Table 10. I would remove that column. |
As suggested, this column has been removed. Thank you very much for the insightful feedback that improved our article. |
|
The source code is not listed, which is problematic as other research groups cannot reproduce the results. |
Thank you very much for the comment. The primary source code has been included in section 4 (line 531-532) |
|
The TB-DRC-DSS is not where to find and having a URL would be helpful. |
Line 631 now has the URL of the web application.
The URL for the online application is https://itbru.com/TDRC/index.html. |

Reviewer 2 Report
The present study presents an ensemble Deep Learning method to detect tuberculosis and perform X-Ray image classification among 5 classes. The methodology is very clear and each step (segmentation, augmentation etc.) is well-justified. The results are very promising compared to the selected literature. Besides, the authors propose a basic web-based application to assist radiologists. There are some improvements to be made:
· In the related work section, the authors need to determine the state-of-the-art with reference to the reported accuracy. In the current state, this sections only refers to methodology and no results are presented. Though table 3 summarizes the literature results, it would be good to at least inform the reader about the expected outcomes in terms of accuracy metrics of the study in the related work section. For example, table 3 reports an AUC that varies between 0.67 and 0.93. What would an acceptable AUC be in the present study?
· Please include information regarding the golden truth of the used datasets. Is the reference label determined by experts or by any other diagnostic test? If the labels are assigned by the human experts, then it would be better to inform the readers that the current study measures the agreement between the developed AI model and the experts. If not, the model performs diagnosis.
· The authors do not use a proper k-fold cross validation for training-testing splits. Random train-test split once is not preferable in such tasks, especially when the datasets are not of very large scale. This is a severe limitation of the study. Please, include this in the comparison tables and mention the evaluation method of the compared studies. Otherwise, I would strongly recommend the authors to re-evaluate the models with k-fold cross validation.
· Table 14 showcases some extra results. However, the selected test set of 428 images come from the training set. If not, please clarify. If the test set comes from the train set, there is no point in presenting this result.
· The authors should consider employing post-hoc explainability methods (e.g. LIME, Grad-CAM) for providing explanations. This is a limitation of the particular study.
Some minor changes:
· Please increase the number of references in the first three paragraphs of the Introduction. For example “A person…has it”: it is not supported by any literature.
· The nature of the employed data (e.g. X-Rays) is not mentioned in the Abstract
· Lines 100-102 do not make sense, please correct
· Please correct line 170 “Radionics”
· Figure 1 is not cited in text
· Figure 4: please add “binary mask” near the “output image”
· The methodology is very straightforward. However, section 3.8 needs to be more concise.
· Please correct lines 528-532
· Table 8, column 2: “egmentation” and Table 10 “COVID-10”
Author Response
Thank you for your valuable comments. The authors have tried to answers all of your recommendations as the attached file.

Round 2
Reviewer 2 Report
The authors have made substantial improvements to their manuscript and addressed all my comments.